# Graph Neural Networks with Multiple Feature Extraction Paths for Chemical Property Estimation

**DOI:** 10.3390/molecules26113125

**Published:** 2021-05-24

**Authors:** Sho Ishida, Tomo Miyazaki, Yoshihiro Sugaya, Shinichiro Omachi

**Affiliations:** Graduate School of Engineering, Tohoku University, Sendai 9808579, Japan; sho0849@iic.ecei.tohoku.ac.jp (S.I.); sugaya@iic.ecei.tohoku.ac.jp (Y.S.); machi@ecei.tohoku.ac.jp (S.O.)

**Keywords:** chemical property estimation, graph neural networks, molecular data, multiple feature extraction

## Abstract

Feature extraction is essential for chemical property estimation of molecules using machine learning. Recently, graph neural networks have attracted attention for feature extraction from molecules. However, existing methods focus only on specific structural information, such as node relationship. In this paper, we propose a novel graph convolutional neural network that performs feature extraction with simultaneously considering multiple structures. Specifically, we propose feature extraction paths specialized in node, edge, and three-dimensional structures. Moreover, we propose an attention mechanism to aggregate the features extracted by the paths. The attention aggregation enables us to select useful features dynamically. The experimental results showed that the proposed method outperformed previous methods.

## 1. Introduction

Each molecule has its unique chemical properties. Estimation of the chemical properties is the first step in the field of drug discovery. Reagent testing is a standard estimation method. However, its process requires long time and equipment cost. Machine learning methods have been widely studied to reduce time and cost.

Most machine learning methods transform molecules into feature vectors and estimate chemical properties using a neural network. There is a high correlation between molecular structure and chemical properties. For example, molecules with benzene rings have a sweet aroma and flammability, and hydroxy groups (OH groups) are readily soluble in water. Therefore, feature extraction of molecular structures is essential in the estimation of chemical properties using machine learning. Since chemical properties depend on an essential structure, a flexible feature extraction method is necessary. A general feature extraction method is Molecular Fingerprints [1,2,3,4], which transform a molecular structure into a one-hot vector of the presence or absence of specific structures designed by humans. However, the specific structures are hard for modification according to chemical properties since experts need to change the specific structure of Molecular Fingerprints.

Recently, feature extraction using graph convolutional neural networks [5] has been attracting attention as a learnable feature extraction method. As shown in Figure 1, the graph represents the molecule using nodes (atoms) and edges (bonds). Node features are extracted by updating their features and neighboring node features. The node feature propagates to further nodes by the number of update processes. Besides, the update is based on a neural network. Thus, a feature extraction model can learn the essential substructures in a molecule according to the chemical characteristics of the estimation target. Various models using graph convolutional neural networks have been developed [6,7]. The weave model [6] extracts edge features to consider relationships between nodes. The 3DGCN model used relative coordinates between nodes to extract features of three-dimensional structures [7].

Graph convolutional neural networks worked well in a classification problem, such as active or inactive. However, there is room for improvement in the regression problem due to its extensive estimation range. Furthermore, substructures can be different for target properties. Thus, it is essential to consider multiple structural features simultaneously.

In this paper, we propose a method for chemical property estimation of molecules using multiple structural features. Specifically, we integrate feature extraction paths that consider nodes, edges, and three-dimensional structures, respectively. For more flexible feature extraction, we utilize an attention mechanism to select useful features dynamically.

## 2. Related Work

In the estimation of chemical properties by machine learning, the estimator uses feature vectors extracted from molecules. Molecular Fingerprint is a method for extracting feature vectors from molecules [1,2,3]. This method uses a one-hot vector to represent the presence or absence of human-designed molecular structures. An improved method is Extended Connectivity Fingerprints, or ECFP [4]. ECFP extracts the presence or absence of subgraphs within molecular radius as a feature vector. However, these methods only consider pre-designed molecular structures and, thus, cannot extract features according to the chemical properties of the target.

A flexible feature extraction method has been developed using machine learning. Duvenaud et al. used neural networks to refine the features of ECFP [8]. Recently, the graph convolutional neural network [5] has attracted much attention. Graph convolutional neural networks sequentially update node features using the features of their neighborhood nodes. Finally, all the node features are merged into a one-dimensional feature vector, resulting in a feature vector of a molecule. In addition to estimating molecular properties, graph convolutional neural networks are used in a wide range of fields, including language processing [9,10,11], human motion estimation [12,13,14], graph similarity estimation [15,16], and class identification [17,18].

For the estimation of chemical properties, various models exist [19,20,21,22,23,24]. Directed graphs are used to reduce computation and update node features [22,23]. Edge features are extracted in References [6,25]. Relative coordinates between nodes are used to extract features of three-dimensional structures [7]. There are methods that learn the importance of node features [26,27]. However, the aforementioned methods specialized in a specific molecular structure, such as edge and three-dimensional structures. In this study, we propose to integrate three feature extraction methods [5,6,7] to simultaneously extract multiple molecular structures. Furthermore, we dynamically select features using an attention mechanism to improve the estimation performance.

## 3. Materials and Methods

We propose a graph convolutional neural network that integrates three different approaches of feature extraction. Depending on the chemical properties of the estimation target, we need to extract different features. Therefore, we simultaneously extract node features, edge features, and three-dimensional features. Furthermore, we use attention to calculate the importance of each feature dynamically. As shown in Figure 2, the proposed method extracts features using multiple paths (node features, edge features, and three-dimensional features) and aggregates each feature. Firstly, we extract features through each path. Then, we form a molecular feature by aggregating the features. The proposed method enables us to consider various structures of the molecule by extracting features through multiple paths.

### 3.1. Node Feature Extraction Path

This path extracts node features using relationships between nodes. Let $H_{i}^{t}\in R^{M\times 1}$ represents a *M*-dimensional feature vector of node *i* at *t*-th update round. We produce the pair feature $P_{ij}\in R^{M\times 1}$ between node *i* and *j* by Equation (Equation 1), where σ represents the rectified linear unit, and ‖ is the concatenation operation. A weight $W_{\mathrm{np}}\in R^{M\times 2M}$ and a bias $B_{\mathrm{np}}\in R^{M\times 1}$ are learning parameters. Subsequently, we update the node features $H_{i}$ as Equation (2). N(i) is the set of neighboring nodes of node *i*. The weight is $W_{n}\in R^{M\times M}$, and the bias is $B_{n}\in R^{M\times 1}$.
(1)$$\begin{array}{cc} P_{ij} & =\sigma (W_{\mathrm{np}}\left(H_{i}^{t}\|H_{j}^{t}\right)+B_{\mathrm{np}}), \end{array}$$
(2)$$\begin{array}{cc} H_{i}^{t+1} & =\sigma (\sum_{j\in N(i)}W_{n}P_{ij}+B_{n}). \end{array}$$

### 3.2. Edge Feature Extraction Path

The extraction path of edge features takes into account the edge relationships between nodes. The atoms in a molecule can have various bonds, such as single bonds and double bonds. We incorporate these bond types into the feature extraction to consider the molecular structure’s connectivity.

We use the five bonds: single bonds, double bonds, triple bonds, aromatic bonds, and bonds to themselves. Let *N* represent the number of atoms. We represent the bonds using the edge parameter $E\in R^{N\times N}$ to describe the connectivity types. A naive parameter for *E* is using categorical values, such as 1 for the single bond. Inspired by Reference [25], we learn parameters to represent the bonds rather than merely representing the bonds using five categorical values. As shown in Figure 3, we create five adjacency matrices and learn the edge parameters using convolutional filter. The convolution filter has a kernel size of 1, and the number of channels is 5.

We obtain a pair feature $P_{ij}$ by Equation (Equation 3) using $E_{ij}$, the (i,j)th element of *E*. Note that $E_{ij}$ is a scalar value. Then, we update the feature $H_{i}$ by Equation (4). The learning parameters are weight $W_{\mathrm{ep}}\in R^{M\times 2M},W_{e}\in R^{M\times M}$, bias $B_{\mathrm{ep}},B_{e}\in R^{M\times 1}$. We take the molecular bonds into account by multiplying *E* and the paired features.
(3)$$\begin{array}{cc} P_{ij} & =\sigma (E_{ij}W_{\mathrm{ep}}\left(H_{i}^{t}\|H_{j}^{t}\right)+B_{\mathrm{ep}}), \end{array}$$
(4)$$\begin{array}{cc} H_{i}^{t+1} & =\sigma (\sum_{j\in N(i)}W_{e}P_{ij}+B_{e}). \end{array}$$

### 3.3. Three-Dimensional Feature Extraction Path

We incorporate three-dimensional structural information into feature updates based on Reference [7]. Let $\left(x_{i},y_{i},z_{i}\right)$ represent the absolute coordinate of node *i*, we calculate the relative coordinate $R{\left(x\right)}_{ij}=x_{i}-x_{j}$ of the *x*-coordinate. Likewise, we obtain relative coordinate in *y* and *z*, $R{\left(y\right)}_{ij}=y_{i}-y_{j}$ and $R{\left(z\right)}_{ij}=z_{i}-z_{j}$.

We calculated the pair feature $P_{ij}$ using the relative coordinates *R* as defined in Equation (Equation 5). Then, we obtain the intermediate feature $Q_{i}$ by accumulating the pair features as in Equation (6). However, $Q_{i}$ exclude node feature $H_{i}$ due to $R_{ii}=0$. Therefore, as shown in Equation (7), we propose to concatenate $H_{i}$ and $Q_{i}$.
(5)$$\begin{array}{cc} P_{ij} & =\sigma (\sum_{k\in (x,y,z)}R{\left(k\right)}_{ij}W_{\mathrm{tp}}\left(H_{i}^{t}\|H_{j}^{t}\right)+B_{\mathrm{tp}}), \end{array}$$
(6)$$\begin{array}{cc} Q_{i} & =\sigma (\sum_{j\in N(i)}W_{\mathrm{tq}}P_{ij}+B_{\mathrm{tq}}), \end{array}$$
(7)$$\begin{array}{cc} H_{i}^{t+1} & =\sigma (W_{t}\left(H_{i}^{t}\|Q_{i}\right)). \end{array}$$

There is a drawback in relative coordinates. The difference of relative coordinates is affected by translation and rotation. For further improvements, it is promising to use distance between atoms.

### 3.4. Feature Aggregation

We propose to extract more useful features by merging the features extracted through the paths. We integrate the three features using attention to dynamically select important features for each node. We integrate the features as Equation (Equation 8), where $H^{\mathrm{node}}$, $H^{\mathrm{edge}}$, and $H^{3}d$ represent features extracted by the paths. Where $\alpha_{i}^{\mathrm{node}}$ represents an attention for $H_{i}^{\mathrm{node}}$ at node *i*, which is defined in Equation (9). We used the softmax function to obtain α. Inspired by Reference [26], we calculate $e_{i}^{\mathrm{node}},e_{i}^{\mathrm{edge}},e_{i}^{3}d$ for each feature by Equation (10). We use the initial feature $H_{i}^{\mathrm{init}}$ of the node *i* and $H^{p},p\in \left\{\mathrm{node},\mathrm{edge},3d\right\}$.
(8)$$\begin{array}{cc} H_{i} & =\sigma (W_{\mathrm{agg}}\sum_{p\in \{\mathrm{node},\mathrm{edge},3d\}}\alpha_{i}^{p}H_{i}^{p}), \end{array}$$
(9)$$\begin{array}{cc} \alpha_{i}^{p} & =\mathrm{softmax}\left(e_{i}^{p}\right)=\frac{exp(e_{i}^{p})}{\sum_{k\in \{\mathrm{node},\mathrm{edge},3d\}}exp\left(e_{i}^{k}\right)}, \end{array}$$
(10)$$\begin{array}{cc} e_{i}^{k} & =W_{\mathrm{att}}\left(\sigma \left(H_{i}^{\mathrm{init}}W_{\mathrm{init}}\right)\|\sigma \left(H_{i}^{k}W_{k}\right)\right). \end{array}$$

### 3.5. Details of the Proposed Model

We illustrated the structure of the proposed model in Figure 4. The proposed model extracts features using the paths and aggregation, which are composed of graph convolutional neural networks. Then, we sum up the features along to each dimension to produce a molecular feature vector. Finally, we estimate chemical properties by applying a fully connected layer.

We adopted two-stage training. Specifically, we independently trained each path. Then, we fixed the paths and trained the aggregation layer and the fully connected layer. We used the mean square error (MSE) loss for training. We followed Reference [7] to determine initial features, resulting in 60 dimensions feature vectors. The batch size was set to 16.

### 3.6. Datasets and Metrics

We mainly used two datasets in the experiments: Freesolv and ESOL. Each of these datasets has been compiled in Reference [28] and is widely used as a dataset to evaluate methods for estimating chemical properties. Freesolv is a dataset for estimating the free energy of hydration of molecules and contains 1128 molecules. ESOL is a dataset for estimating solubility and contains 643 molecules. Overall, Freesolv and ESOL are regression task, which directly predicts the values. Besides, we used four datasets for verification of the proposed method. We summarized the datasets in Table 1. QM8 has four excited state properties calculated by three different methods. Thus, 12 properties in total.

We randomly split the dataset to 8:1:1 for training data, validation data, and test data. We evaluated the proposed method and the comparison methods for 10 trials. We calculated the average of the metrics over the trials. The evaluation metrics is Mean Absolute Error (MAE). The smaller MAE is better.

## 4. Results

### 4.1. Comparison Methods

As comparison methods, we used the graph convolutional neural network (GCN) [5], the Weave model [6], and the 3DGCN [7]. Broadly, the comparison methods extract node features (GCN), edge features (Weave), and three-dimensional features (3DGCN), respectively. We set the number of updating layers to two in the proposed method and the comparison methods for equivalent comparison. In addition, the summation is used for producing molecular features as same as the proposed method. The main difference between the comparison methods and the proposed method is the number of feature extraction paths. The comparison methods have a single path. In contrast, the proposed method has multiple paths to consider node, edge, and three-dimensional structure simultaneously.

### 4.2. Main Results

We trained models until they converged. We stopped training if the loss is no longer improving for ten successive epochs. We defined no improvement if improvements are less than 0.0001. Figure 5 shows typical loss curves of the proposed method in training and validation. The loss curves of the validation also converged. Thus, there was no overfitting. The models successfully converged. The proposed model has 143,286 parameters. Compared to 135 M and 11.4 M parameters in VGG-16 and ResNet-18 models, the number of parameters is significantly small. Therefore, the numbers of data points in Freesolv and ESOL are satisfactory for the proposed method.

We showed the numerical results in Table 2. GCN was better among the comparison method. Moreover, the proposed method outperformed the comparison methods on all datasets. The proposed method successfully learned the essential features. Thus, the results showed the effectiveness of the multiple feature extraction for chemical property regression.

### 4.3. Results on Quantum Mechanism

We conducted experiments using QM7 and QM8 datasets. We trained models until they converged. The results are shown in Table 3. The results show that the proposed method outperformed the comparison methods at ten tasks. In addition, the proposed method was the second-best on the rest tasks. Thus, we verified the effectiveness of the proposed method in various tasks.

### 4.4. Evaluation on Aggregation Approaches

We carried out experiments to discuss the effectiveness of the feature aggregation. Besides the attention, there can be various aggregation approaches, such as concatenation, summing, and maximum. We define them in Equations (Equation 11)–(13).
(11)$$\begin{array}{cc} H_{i} & =\sigma (W_{\mathrm{concat}}(H_{i}^{\mathrm{node}}\|H_{i}^{\mathrm{edge}}\|H_{i}^{3}d)), \end{array}$$
(12)$$\begin{array}{cc} H_{i} & =\sigma (W_{\mathrm{sum}}\left(H_{i}^{\mathrm{node}}+H_{i}^{\mathrm{edge}}+H_{i}^{3}d\right)), \end{array}$$
(13)$$\begin{array}{cc} H_{i} & =\sigma (W_{\max}\max\left(H_{i}^{\mathrm{node}},H_{i}^{\mathrm{edge}},H_{i}^{3}d\right)). \end{array}$$

Table 4 shows the results. The attention and the concatenation aggregations were the best on Freesolv and ESOL, respectively. All the aggregations achieved accurate estimation results. Thus, the proposed method has capability to aggregate different features using various approaches.

### 4.5. Impacts of Feature Extraction Paths

We conducted experiments to clarify the impacts of feature extraction paths on the datasets. We built models with one-, and two-paths. Specifically, one-path models have a single path of node, edge, and three-dimensional features, respectively. The two-path models have node and edge paths, node and three-dimensional paths, and edge and three-dimensional paths, respectively. We used the attention aggregation in the two-path models.

The results are shown in Table 5. In Freesolv, the two-path models outperformed the one-path models. Furthermore, the proposed model was the best at 0.639. Likewise, two-path models were superior to the one-path models on ESOL. Overall, multiple features were significant in chemical property estimation.

### 4.6. Results in Classification Tasks

We conducted classification experiments on BACE and BBBP. In addition, we trained models until they converged. We used four metrics: Accuracy, Recall, Precision, and F-score. Table 6 and Table 7 show the results. The two path model of the proposed method achieved the best results on BACE at all metrics. In addition, the proposed method was the best at precision and the second-best at the other metrics on BBBP. The ROC curves in Figure 6 shows the significant performance of the proposed method. These results show that the effectiveness of the proposed method in a classification task.

### 4.7. Verification of Edge Parameters

We carried out experiments to verify the effect of edge parameter *E*, a learning parameter in Equation (Equation 3). We compared two modes on the basis of the edge path model. One used edge parameters learned by convolution with kernel sizes. The other adopted the fixed edge parameter, e.g., categorical values, one for the self node, 2 for single bonds, 3 for double bonds, 4 for triple bonds, 5 for aromatic bonds. We adopted 50 epochs for training. The results are shown in Table 8. The model using the learned edge parameters improved on all the datasets. Therefore, the proposed method learned an optimal representation of edge types.

### 4.8. Effect of the Self Node in Three-Dimensional Features

We evaluated the effect of the self node in the three-dimensional feature. We used the model of the extraction path of three-dimensional features. Then, we compared the model with and without the self node. Specifically, we defined the model without self node as Equation (Equation 14) instead of Equation (7).
(14)$$H_{i}^{3}d=\sigma \left(W_{\mathrm{tq}}Q_{i}\right).$$

If we omit the self nodes, the self nodes were not considered when aggregating the pair features and updating the node features. The experimental results are shown in Table 9. There were specific improvements by the self node. Therefore, we confirmed that the performance could improve by incorporating the self nodes into three-dimensional features.

### 4.9. Attention Visualization

We conducted experiments to confirm the capability of dynamic determination for the attention values in the proposed method by visualizing each node’s attentions $\alpha_{i}$. According to Equation (9), the summation of $\alpha_{i}$ among the paths is normalized to one. Thus, we directly illustrated α using bar charts. The visualization results are shown in Figure 7. The various attention values were assigned to each node. The result shows that the proposed method flexibly determined attention for each node.

## 5. Conclusions

In this study, we proposed a method for chemical property estimation in molecules. The proposed method uses multiple paths to extract features focusing on specific structures, such as node relationship, edge relationship, and three-dimensional structure in a molecule. Furthermore, we proposed to obtain more useful features by aggregating multiple features by selecting essential features dynamically. Compared to existing methods that focus on only one structure, the experimental results showed that the proposed method outperformed the comparison methods in regression tasks. Therefore, multiple feature extraction can improve the performance of chemical property estimation in molecules.

## Figures and Tables

**Figure 1 molecules-26-03125-f001:**
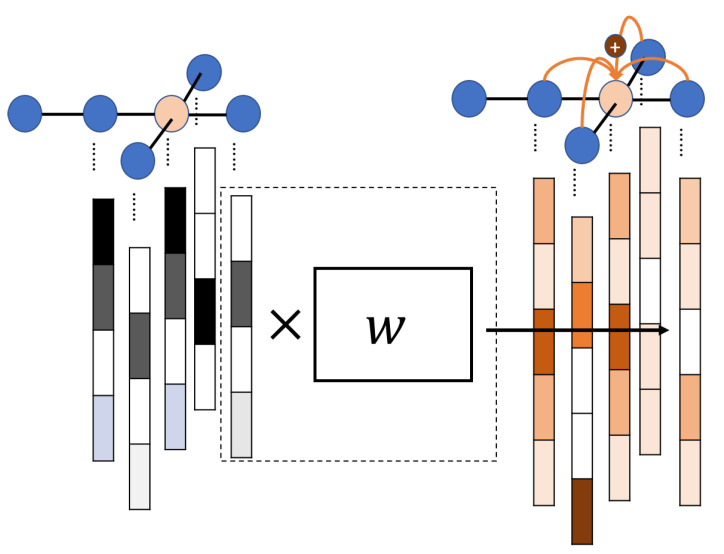
Graph convolutional neural network. Node features are updated with weights *w* (node features after arrows in the figure). Then, the node of interest (orange) is updated with the features of its neighbors.

**Figure 2 molecules-26-03125-f002:**
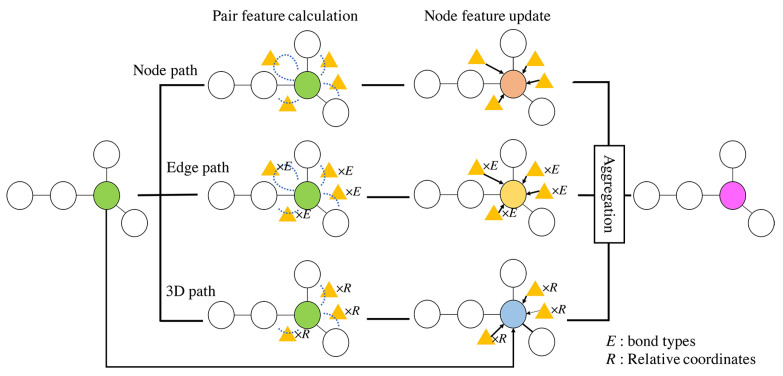
Overview of the proposed method. For simplicity, we illustrated the flow of feature update for a single attention node (green circle). Firstly, we generate pair features (yellow triangles) representing the relationship between the node and its neighbors. We use bond types and relative coordinates to extract edge relationships and three-dimensional structures. Then, we update the node features using the pair features (orange, yellow, and blue circles). Finally, we aggregate each path’s features of the attention node to obtain the node features, which are the output of this layer (purple circles). We repeat the above processes for feature updates.

**Figure 3 molecules-26-03125-f003:**
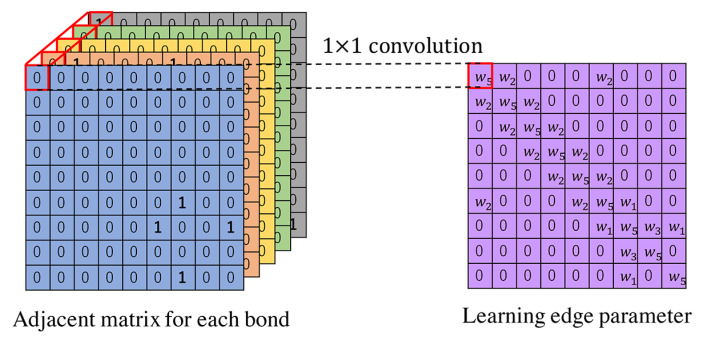
Edge parameters.

**Figure 4 molecules-26-03125-f004:**
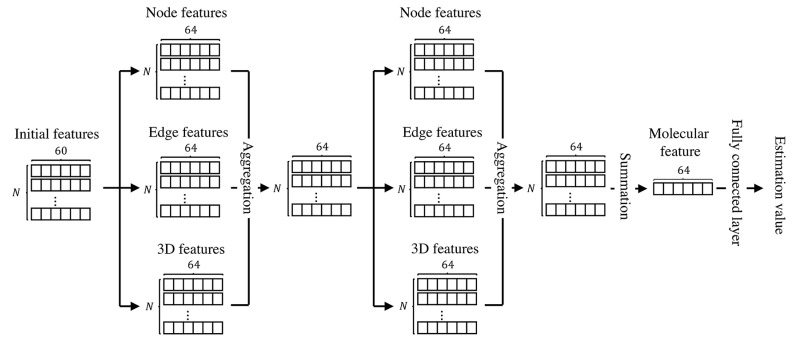
The structure of the model. The initial features are the 60-dimensional features.

**Figure 5 molecules-26-03125-f005:**
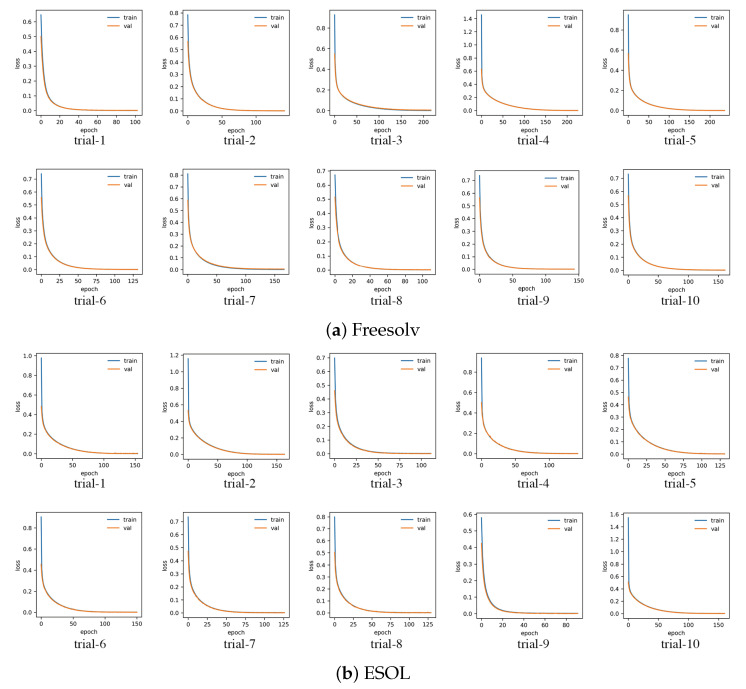
Loss curves of the proposed model.

**Figure 6 molecules-26-03125-f006:**
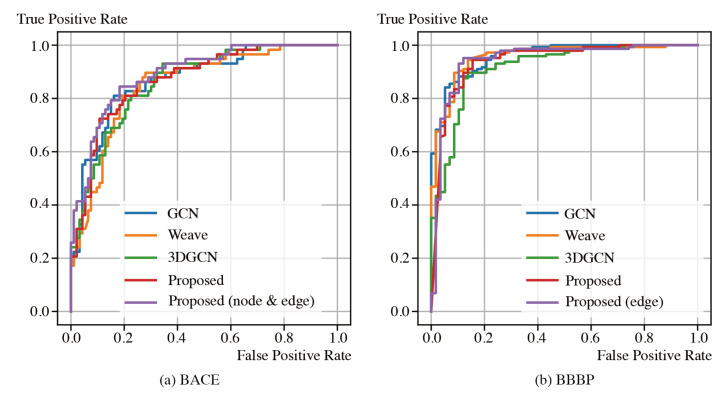
ROC curves.

**Figure 7 molecules-26-03125-f007:**
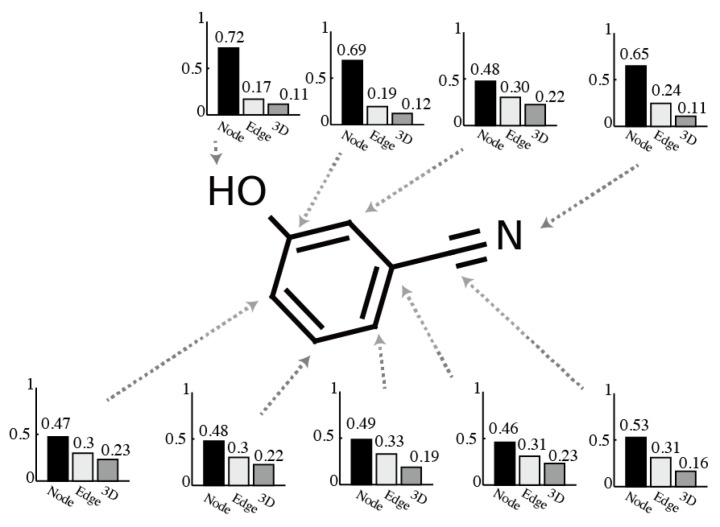
Visualization results of attention values.

**Table 1 molecules-26-03125-t001:** Summary of the dataset used in the experiments.

Dataset	#Mols	Category	Task
Freesolv	1128	Physical chemistry	Regression for water solubility
ESOL	643	Physical chemistry	Regression for hydration free energy
QM7	7160	Quantum mechanism	Regression for Atomization energy
QM8	21,786	Quantum mechanism	Regression for excited state properties
BACE	1513	Biophysics	Classification for inhibitors of β-secretase 1
BBBP	2039	Physiology	Classification for blood-brain barrier penetration

**Table 2 molecules-26-03125-t002:** Averages of MAE over 10 trials (Bold and underline are the best and the second-best, respectively).

	GCN	Weave	3DGCN	Proposed
Freesolv	0.764	0.817	0.743	**0.717**
ESOL	0.503	0.665	0.531	**0.498**

**Table 3 molecules-26-03125-t003:** Averages of MAE over 10 trials on quantum mechanism datasets (Bold and underline are the best and the second-best, respectively).

	GCN	Weave	3DGCN	Proposed
QM7	10.75	12.22	12.89	**9.13**
QM8 (E1-CC2)	0.00846	**0.00608**	0.00651	0.00611
QM8 (E2-CC2)	0.0099	0.0080	0.0081	**0.0077**
QM8 (f1-CC2)	0.0180	0.0161	0.0148	**0.0141**
QM8 (f2-CC2)	0.0346	0.0327	0.0324	**0.0303**
QM8 (E1-PBE0)	0.0082	**0.0064**	0.0071	0.0066
QM8 (E2-PBE0)	0.00945	**0.00705**	0.00822	0.00712
QM8 (f1-PBE0)	0.0154	0.0120	0.0124	**0.0114**
QM8 (f2-PBE0)	0.0291	0.0259	0.0261	**0.0247**
QM8 (E1-CAM)	0.0074	0.0061	0.0065	**0.0058**
QM8 (E2-CAM)	0.0084	0.0065	0.0071	**0.0063**
QM8 (f1-CAM)	0.0166	0.0132	0.0127	**0.0123**
QM8 (f2-CAM)	0.0308	0.0268	0.0275	**0.0259**

**Table 4 molecules-26-03125-t004:** Averages of MAE for aggregation approaches.

	Concat	Sum	Max	Attention
Freesolv	0.666	0.663	0.703	**0.639**
ESOL	**0.472**	0.478	0.488	0.484

**Table 5 molecules-26-03125-t005:** Average of MAE for path combinations.

	Path Combinations						
Node	✓			✓	✓		✓
Edge		✓		✓		✓	✓
3D			✓		✓	✓	✓
Freesolv	0.710	0.702	0.864	0.640	0.676	0.685	**0.639**
ESOL	0.483	0.498	0.538	0.477	**0.476**	0.482	0.484

**Table 6 molecules-26-03125-t006:** Classification results on BACE (Bold and underline are the best and the second-best, respectively).

	GCN	Weave	3DGCN	Proposed	Proposed (Node & Edge)
Accuracy	0.807	0.751	0.774	0.799	**0.811**
Recall	0.779	0.714	0.726	**0.782**	**0.782**
Precision	0.777	0.739	0.749	0.763	**0.783**
F-score	0.778	0.726	0.737	0.773	**0.782**

**Table 7 molecules-26-03125-t007:** Classification results on BBBP (Bold and underline are the best and the second-best, respectively).

	GCN	Weave	3DGCN	Proposed	Proposed (Edge)
Accuracy	**0.886**	0.871	0.873	0.874	0.884
Recall	**0.942**	0.915	0.923	0.922	0.928
Precision	0.912	0.917	0.912	0.915	**0.922**
F-score	**0.927**	0.916	0.918	0.918	0.925

**Table 8 molecules-26-03125-t008:** Averages of MAE using fixed and learned edge parameters.

	Fixed	Conv-1	Conv-3	Conv-5	Conv-7
Freesolv	1.181	**0.799**	0.872	0.957	0.927
ESOL	0.706	**0.525**	0.598	0.585	0.654

**Table 9 molecules-26-03125-t009:** Aberage of MAE with and without self node features.

	w/o Self	w/ Self
Freesolv	1.525	**0.864**
ESOL	0.726	**0.538**

## Data Availability

The datasets are available at http://moleculenet.ai/datasets-1 (accessed on 14 May 2021).

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
