# Peer review of "Graph Neural Networks with Multiple Feature Extraction Paths for Chemical Property Estimation"

_molecules, 2021, doi:10.3390/molecules26113125_

Round 1

Reviewer 1 Report

MS:
Title:
Molecules-1152974
Graph Neural Networks with Multiple Feature Extraction Paths for Chemical Property
Estimation

Authors: Sho Ishida, Tomo Miyazaki, Yoshihiro Sugaya and Shinichiro Omachi

The authors describe a method to combine 3 different features for representing molecules for machine/deep learning tasks utilizing convolutional neural networks. The features are extracted as node, edge and 3D features, which are updated by backpropagation to minimize loss. The potential of the new approach introduced in this manuscript is high. A strict validation of the method could significantly reach out to a large audience in chemistry, biology and computer science. Therefore, this referee recommends publication after consideration of the following points: 

Major comments: 

1) The convolutional kernel used was of size 1. A highlight on how the performance changes over various kernel sizes would be helpful. Technically, convolution networks are designed to extract local features by considering the local assembly with kernel sizes greater than 1. Based on the way
features are represented, a larger kernel size could boost the performance.

2) Based on the previous comment; if one would replace the convolutional neural network by dense layers, would the model perform in the same way? The convolutional kernel of 1 can be technically reproduced via dense networks.

3) The 3D features are extracted by taking coordinate differences. Can this be replaced by the distance between the atoms? Differences in coordinates always depend on the choice of coordinates and therefore, could be influenced by the overall translation and rotation of the molecule.

4) The authors report a fixed number of 50 epochs. However, the model should be trained until convergence and should not be restricted over number of epochs. A plot representing the loss over epochs would help the readers understand if the convergence criteria was satisfied.

5) The two datasets Freesolv and ESOL were used for evaluation which contain in total 1128 and 643 data points respectively. Considering the fact that the authors used deep neural networks in combination of convolution neural networks, their dataset are too small. The number of weights to be optimized could turn out to be significantly larger than the number of data points. Additionally, an exact specification of the number of weights would be helpful for the readers.

6) Based on the previous comment, how did the authors make sure that there was no overfitting? A plot of loss over epochs for training, validation/test could help clarify this.

7) Considering larger datasets will lead to a much better verification of the model. One more problem apart from overfitting with the small dataset could be data leakage. The authors should either mention how they remove data leakage or use a larger dataset.

8) The authors should not restrict themselves to prediction of free energy of hydration. Predicting only one property could be an artifact of learning features unrelated to the training tasks. For example, although very complex, maybe the number and types of atoms could influence the hydration energy for the dataset used, unrelated to edges. A large number of datasets is already available for prediction tasks, such as the QM7 dataset that contains energy, HOMO-LUMO gap, zero-point energy etc. for more than 7000 small compounds.

9) The comparison of methods was solely done on the basis of MAE and no classification tasks were performed. A comparison based on classification by comparing Receiver Operator Characteristics (ROC) would be helpful for reaching out to a broader audience.

10) The attention network visualization highlights almost all of the atoms. Given that hydration energies are more involved with polar functional groups, what is the model learning? Perhaps an intuition over high attention atoms would help understanding.

11) Attention networks are complex due to the involvement of neural weight updates that might lead to misleading understanding of the attention. One suggestion would be to use Grad-CAM method to point out exact atoms and bonds in the compound that are most influential for hydration
energies based on the convolution networks.

Minor comments:
Page 2, line 64: Change “vector of a molecular” to “vector of a molecule”
Page 4, line 98: Change “molecular” to “molecule”

Reviewer 2 Report

The manuscript presented covers an interesting topic in med chem. However, some improvement are required for its publication:

-the manuscript should be organized following the guidelines of the journal. Accordingly  section namely Materials and Methods should be included

- the main concern related to the manuscript is the lack of validation of the proposed methods. Authors should clearly validate the method adopting an external set evaluating the performance of the method by ROC curve analysis for example. This step is necessary to obtain a comprehensive assessment of the method.

Round 2

Reviewer 1 Report

The authors have carefully considered all comments/suggestions of this referee, therefore the paper should be published.

Reviewer 2 Report

Concerns were addressed in the revised version